# A Natural CHI3L1—Targeting Compound, Ebractenoid F, Inhibits Lung Cancer Cell Growth and Migration and Induces Apoptosis by Blocking CHI3L1/AKT Signals

**DOI:** 10.3390/molecules28010329

**Published:** 2022-12-31

**Authors:** Da Eun Hong, Ji Eun Yu, Jin Woo Lee, Dong Ju Son, Hee Pom Lee, Yuri Kim, Ju Young Chang, Dong Won Lee, Won Kyu Lee, Jaesuk Yun, Sang Bae Han, Bang Yeon Hwang, Jin Tae Hong

**Affiliations:** 1Medical Research Center, College of Pharmacy, Chungbuk National University, Osongsaengmyeong 1-ro, 194-21, Osong-eup, Heungduk-gu, Cheongju 28160, Republic of Korea; 2Department of New Drug Development Center, Osong Medical Innovation Foundation (KBio Health), Cheongju 28644, Republic of Korea

**Keywords:** ebractenoid F, CHI3L1, AKT, lung cancer cell growth inhibition

## Abstract

Our previous big data analyses reported a strong association between CHI3L1 expression and lung tumor development. In this present study, we investigated whether a CHI3L1-inhibiting natural compound, ebractenoid F, inhibits lung cancer cell growth and migration and induces apoptosis. Ebractenoid F concentration-dependently (0, 17, 35, 70 µM) and significantly inhibited the proliferation and migration of A549 and H460 lung cancer cells and induced apoptosis. In the mechanism study, we found that ebractenoid F bound to CHI3L1 and suppressed CHI3L1-associated AKT signaling. Combined treatment with an AKT inhibitor, LY294002, and ebractenoid F synergistically decreased the expression of CHI3L1. Moreover, the combination treatment further inhibited the growth and migration of lung cancer cells and further induced apoptosis, as well as the expression levels of apoptosis-related proteins. Thus, our data demonstrate that ebractenoid F may serve as a potential anti-lung cancer compound targeting CHI3L1-associated AKT signaling.

## 1. Introduction

Chitinase-3 like 1 (CHI3L1), of the glycoside hydrolase 18 (GH18) family of chitinases, is secreted by macrophages, neutrophils, synoviocytes, chondrocytes, epithelial cells, and smooth muscle cells [1,2,3]. Many human cancers, such as breast, colon, prostate, lung, and liver cancers, have been reported to show increased CHI3L1 expression [1,4,5,6]. Both non-small-cell lung carcinoma (NSCLC) and small-cell lung carcinoma (SCLC) patients have been reported to show high CHI3L1 expression and serum levels, and these have been reported to be associated with poorer overall survival. High CHI3L1 expression and serum levels have been found in both non-small-cell lung carcinoma (NSCLC) and small-cell lung carcinoma (SCLC) patients, and they have been linked to worse overall survival, [7] a poorer response to chemotherapy, and worse chemotherapy responses [8]. In our previous study, we also found that CHI3L1 is markedly associated with the development of lung cancer [8].

CHI3L1 has been reported to activate its downstream signaling pathways, such as protein kinase B, also known as AKT, and Wnt/β-catenin, as well as MMP signaling pathways, which have been reported to be significantly associated with lung cancer cell growth [9]. Further, it was reported that CHI3L1 activates the AKT pathway in melanoma metastasis [9]. It is also noteworthy that CHI3L1 signaling promotes oxidant-induced apoptosis, lung injury, and melanoma metastasis through activation of the AKT pathway [10]. In our previous study, we found that CHI3L1-inhibiting compound K284 inhibited lung tumor metastasis and lung cancer cell growth via the inhibition of AKT signaling [8]. Moreover, in cells with CHI3L1 deletion, the suppression effects of K284 on the AKT signaling, cancer cell growth and migration, and apoptotic cell death were abolished [11].

Despite extensive evidence regarding the role of CHI3L1 and its downstream signals in lung tumor growth, knowledge regarding compounds disrupting CHI3L1-specific downstream signaling, and consequently, preventing lung cancer growth, is limited. Herein, we screened several natural compounds using a luciferase activity assay, and selected ebractenoid F as a candidate compound, because it showed strong inhibitory activity against CHI3L1 and has notable cytotoxic effects against lung cancer cells. Ebractenoid F is one of the active ingredients of Euphorbia ebracteolata. The roots of E. ebracteolata are well known as traditional drugs for treating chronic inflammation, including bronchitis, tuberculosis, and psoriasis [12]. A study has shown that ebractenoid F causes anti-inflammatory effects in macrophages through the reduction in nitrogen monoxide in mouse macrophages at non-toxic doses [13]. In addition, the specific mechanism of pharmacological efficacy is not known, so this study reveals the mechanism of ebractenoid F. Moreover, this compound has good drug properties. Thus, in the present study, we investigated whether ebractenoid F inhibits lung cancer cell growth and migration, as well as whether it induces apoptosis through disruption of the CHI3L1/AKT signals.

## 2. Results

### 2.1. Ebractenoid F Was Selected as a Useful Substance Targeting CHI3L1

We screened 58 natural compounds, of which we selected ebractenoid F, using the cell viability assay and CHI3L1 luciferase assay. Ebractenoid F (#.31) effectively inhibited A549 lung cancer cell growth (Appendix A). The transcriptional activities of CHI3L1, as assessed using the luciferase assay, were also significantly reduced by ebractenoid F in A549 cells (Appendix A). We then obtained the preliminary absorption, distribution, metabolism, excretion, and toxicity (ADME/toxicity) data for ebractenoid F using in silico computer analysis and determined its druggability. The data showed that ebractenoid F is excellent as a drug, with complementing lipophilic properties and solubility (Table 1). 

### 2.2. Ebractenoid F Inhibited Lung Cancer Cell Growth

We first studied the concentration-dependent effect of ebractenoid F (0, 17, 35, 70 µM) on lung cancer cell proliferation, which was analyzed using the MTT assay. We also found that ebractenoid F inhibited lung cancer cell growth in a concentration-dependent manner, with an IC_50_ value of 60 µM in A549 cells and 54 µM in H460 cells for 24 h, respectively. At 72 h of treatment with ebractenoid F, the IC_50_ value was approximately 38 µM in A549 cells and H460 cells (Figure 1A,C). Lung cancer cells gradually decreased in size and changed into a small round shape and detached on treatment with ebractenoid F (Figure 1B,D). We also found that ebractenoid F inhibited the proliferation of various cancer cells, but to a lesser extent than that observed in lung cancer cells (Appendix A). These data indicated that ebractenoid F inhibited lung cancer cell growth in a concentration-dependent manner.

### 2.3. Ebractenoid F Suppressed the Migration of Lung Cancer Cells

To evaluate the inhibitory effect of ebractenoid F on the migration of lung cancer cells, we conducted a cell migration assay. We evaluated the migrating ability of A549 and H460 cells treated with various concentrations of ebractenoid F using the wound-healing assay. We found that ebractenoid F suppressed the migration of A549 and H460 cells in a concentration-dependent manner (Figure 2A,B). Similarly, cell invasion was determined using the trans-well assay. Ebractenoid F was found to suppress cell invasion in a concentration-dependent manner (Figure 2C,D). Western blot results showed that the expression of migration proteins was significantly and consistently inhibited by ebractenoid F in a concentration-dependent manner (Figure 2E,F). These results suggest that ebractenoid F treatment can inhibit the cell migration of lung cancer cells in a concentration-dependent manner.

### 2.4. Ebractenoid F Arrested the Cell Cycle of Lung Cancer Cells

To determine the cell cycle stage affected by ebractenoid F, A549 and H460 lung cancer cells were treated with different concentrations of ebractenoid F for 24 h; then, the proportion of cells in each phase of the cell cycle was measured using flow cytometry system. When treated with 70 µM of ebractenoid F for 24 h, 10-fold and 20-fold increases in the proportion of A549 and H460 cells, respectively, in the sub-G1 ratio were observed, compared with cells in the respective control groups (Figure 3A,B). The cell cycle was regulated according to the activity of cyclin and the phosphorylation enzymes CDK. A Western blot analysis was performed to determine the expression of signal proteins related to the cell cycle, and ebractenoid F (0, 17, 35, 70 µM) was found to decrease CDK3, Cyclin C, PCNA, CDK4, CDK6, Cyclin D1, CDK2, and Cyclin E expression in a concentration-dependent manner in both A549 and H460 cells (Figure 3C,D).

### 2.5. Ebractenoid F Induced Apoptosis in Lung Cancer Cells

We performed a TUNEL staining assay to determine whether the inhibition of cell growth by ebractenoid F was due to the inhibition of apoptosis, and we used DAPI to stain the nucleus and then analyzed labeled cells with fluorescence microscopy. Consistent with cell growth inhibitory effects, apoptotic cell death was significantly increased in ebractenoid F-treated A549 and H460 cells. The number of apoptotic (DAPI-positive TUNEL-stained cells) A549 and H460 lung cancer cells increased to approximately 23% and 40%, respectively, at an ebractenoid F concentration of 70 μM (Figure 4A,B). The expression of proteins related to cell death, including cleaved caspase-9, cleaved caspase-3, Bax, and Bcl-2, was investigated using Western blot. The expression of pro-apoptotic proteins, cleaved caspase-9, cleaved caspase-3, and Bax increased, and that of anti-apoptotic protein Bcl-2, decreased under treatment with ebractenoid F in a concentration-dependent manner (Figure 4C,D).

### 2.6. Ebractenoid F Interacted with CHI3L1

Previous big data analysis studies have shown that CHI3L1 is a target protein close to several cancer outbreaks. In addition, CHI3L1 is associated with cell growth, invasion, and metastasis. To determine whether ebractenoid F binds to CHI3L1 and inhibits lung cancer cell growth by blocking the CHI3L1 signal, we first investigated the expression of CHI3L1 on ebractenoid F treatment. CHI3L1 expression was decreased by ebractenoid F in a concentration-dependent manner (Figure 5A,B). To determine whether ebractenoid F affects CHI3L1 stability, we performed a cycloheximide (CHX) pulse-chase assay. With the time-dependent treatment of CHX, the expression of CHI3L1 decreased, but when treated together with ebractenoid F, the expression of CHI3L1 significantly decreased even further. The treatment with ebractenoid F reduced the half-life of CHI3L1 in lung cancer cells (Appendix A), suggesting that reduced stability by ebractenoid F could be involved in CHI3L1 expression. We then examined the interactions between ebractenoid F and CHI3L1. The interaction of ebractenoid F-Sepharose 4B and 6B beads with lung cancer cells transfected with myc-tag CHI3L1 (cell lysates) was assessed using a pull-down assay. The interaction of ebractenoid F-Sepharose 4B and 6B with myc-tag CHI3L1 was then detected by immunoblotting with anti-myc tag antibody. The results indicated that ebractenoid F interacted with CHI3L1 (Figure 5C). To identify the binding site of ebractenoid F to CHI3L1, we performed computational docking experiments using SwissDock, a web service based on EADock DSS, which enables a computational docking analysis to predict the ability of ebractenoid F to bind to CHI3L1 [14]. In the CHI3L1 protein structure, 12 positions where the small molecule ebractenoid F can bind are indicated by red spheres (Figure 5D). Ebractenoid F can be docked in 14 different forms in the highest binding potential cavity among these 12 positions (Figure 5E). Among the 14 docking forms of ebractenoid F, the most energetically stable model was finally selected. The binding energy delta G value of this model was calculated to be -7.66 kcal/mol (Figure 5F). Amino acids that play an important role in this docking are tryptophan31, tryptophan69, phenylalanine58, asparagine100, and glycine98, and the different colors indicate different binding patterns (Figure 5G).

### 2.7. Combination Effects of Ebractenoid F and CHI3L1 siRNA on Lung Cancer Cell Growth

Combination therapy is a promising strategy that shows the synergistic effects of two or more drugs used to treat lung cancer simultaneously. These approaches potentially provide therapeutic anticancer effects, such as reduction in tumor growth and metastatic potential, the suppression of mitotically active cells, and the induction of apoptosis [15,16]. To further investigate the anticancer effect of ebractenoid F on CHI3L1, lung cancer cells were transfected with CHI3L1 siRNA and treated with ebractenoid F. The combined treatment with CHI3L1 siRNA and ebractenoid F significantly reduced the expression of migration proteins (MMP9 and MMP13) compared with the treatment with ebractenoid F or CHI3L1 siRNA alone, and the wound healing assay also showed similar results (Figure 6A–C). Apoptosis and the expression of cell death-related proteins (cleaved caspase-3 and Bax) significantly increased with the combined treatment compared to the single treatment (Figure 6A,B,D). These data showed that the combination treatment with CHI3L1 siRNA and ebractenoid F induced a synergistic effect and indicated that CHI3L1 plays a major role in the ebractenoid F-induced inhibitory effects on cancer cell growth and migration, as well as in the induction of cell apoptosis. A rescue assay was performed to further confirm whether CHI3L1 is suppressed by ebractenoid F. A549 and H460 cells were treated with ebractenoid F and then transfected with the myc-CHI3L1 plasmid or control plasmid (myc-vector). As a result, proliferation and migration were increased by the treatment with myc-CHI3L1, and these increases were smaller when treatment was carried out in the presence of ebractenoid F. In addition, cell proliferation and migration inhibited by ebractenoid F treatment could be recovered by myc-CHI3L1 transfection. Thus, these results suggest that ebractenoid F plays a role in preventing the proliferation and migration of lung cancer cells by inhibiting the CHI3L1 expression (Appendix A).

### 2.8. Ebractenoid F Downgraded the AKT Signaling Activation

AKT signal is associated with CHI3L1. CHI3L1 regulates apoptosis, pyroptosis, inflammasome activation, oxidative injury, antimicrobial responses, melanoma metastasis, and TGF-1 development through activation of the MAPK, AKT, and Wnt/-catenin signaling pathways [9]. Thus, AKT activity could be altered by the interaction of ebractenoid F with CHI3L1. AKT and related cell signaling-associated proteins (in their phosphorylated forms) were identified via Western blot; the AKT and JNK pathways (among MAPK kinase pathway) were found to be the major participants (Figure 7A,B). To confirm the accuracy of these findings, the cells were treated with the combination of ebractenoid F and inhibitors or inhibitors alone. We found that the expression levels of CHI3L1, p-AKT, and p-JNK markedly decreased with ebractenoid F. The decrease in the expression levels of CHI3L1, p-AKT, and p-JNK was greater with the combination treatment than with the treatment with ebractenoid F or inhibitors alone. Moreover, the combination treatment with ebractenoid F and the AKT inhibitor LY294002 was more effective than the combination treatment with ebractenoid F and the JNK inhibitor SP600125 (Figure 7C,D).

### 2.9. Combination Treatment with Ebractenoid F and AKT Inhibitor on Cell Viability and Apoptosis

To further demonstrate the involvement of CHI3L1 and the AKT signaling pathways in ebractenoid F-induced lung cancer cell growth inhibition and apoptosis, we compared the effects of combination treatment with ebractenoid F and the AKT inhibitor LY294002 and treatment with ebractenoid F or the AKT inhibitor alone. Lung cancer cells were pretreated with the AKT inhibitor 30 min before their treatment with ebractenoid F; then, the expression levels of proteins related to cell growth and apoptosis and CHI3L1 were analyzed (Figure 8A,B). The combination treatment with ebractenoid F and the AKT inhibitor markedly reduced CHI3L1 expression compared with treatment with ebractenoid F or the AKT inhibitor alone in both A549 and H460 cells. Combination treatment with ebractenoid F and the AKT inhibitor also further decreased cell viability (Figure 8C,D) and increased apoptosis (Figure 8E,F).

## 3. Discussion

Our previous GWAS/OMIM/DEG analysis results demonstrated a substantial correlation between CHI3L1 and lung cancer development [8]. Using a structure-based virtual screening approach, we found that ebractenoid F possesses both strong CHI3L1 inhibitory activity and cell proliferation inhibitory effects. Ebractenoid F also significantly inhibited lung cancer cell growth, markedly suppressed the migration ability of lung cancer cells, and induced apoptosis in these cells. Further, in ebractenoid F-treated A549 and H460 human lung cancer cells, the expression levels of cancer growth- and migration-associated proteins decreased, as well as those of cell death-associated proteins (Bax, cleaved caspase-3 and cleaved caspase-9). Thus, these results indicate that ebractenoid F exerts notable anti-lung cancer effects by targeting CHI3L1 [17,18].

CHI3L1 is overexpressed in lung cancer and tumor-associated macrophages, and its high expression is detected in the serum of metastatic lung cancer patients [19,20,21]. These findings imply that CHI3L1 is an important regulator of lung cancer growth. Our previous findings have shown that the CHI3L1-inhibiting compound (K284-6111) binds to CHI3L1, thus, inhibiting CHI3L1 signals as well as lung tumor growth [11]. Lung cancer growth and metastasis were also effectively controlled through STAT6-dependent M2 polarization inhibition by anti-CHI3L1 antibodies [21]. In addition, several other studies have reported that CHI3L1 is an important factor in lung cancer metastasis [7,22,23]. Consistent with these findings, we found that ebractenoid F binds to CHI3L1, as evidenced by the pull-down assay and docking model study. Ebractenoid F may bind to CHI3L1 (G value = −7.66 kcal/mol), which leads to decreased CHI3L1 expression. We also found that CHI3L1 expression decreased over time in ebractenoid F-treated cells. In association with the inhibition of CHI3L1 expression, ebractenoid F showed effective inhibitory effects on cell proliferation and migration and induced cell death. Combinations of well-designed drug treatments are promising strategies because combination therapy has shown improved effectiveness over the single treatment approach by synergistically targeting major pathways or targeting additional pathways [16]. Clinical studies on patients treated with combination therapy have found an increase in the overall survival rate of lung cancer patients [24,25]. The combination treatment with CHI3L1 siRNA and ebractenoid F showed a synergistic effect on cancer cell migration and apoptosis compared with the single treatment with siRNA or ebractenoid F. Thus, these data confirmed that CHI3L1 is an important factor in the cancer cell growth inhibitory effect of ebractenoid F.

Ebractenoid F has an anti-lung cancer effect, although the exact mechanisms by which it affects CHI3L1 and CHI3L1-associated signaling are yet to be elucidated. CHI3L1 downstream signaling pathways can be induced by CHI3L1 expression. CHI3L1 activates the AKT signaling pathway, thereby regulating cancer cell apoptosis and melanoma metastasis [26]. Notably, the CHI3L1/IL-13R2 (its receptor) complex and transmembrane protein 219 (TMEM219) exacerbate lung damage, melanoma metastasis, and oxidant-induced apoptosis by activating the AKT pathway [27]. CHI3L1 is critical in the activation of the AKT pathway, and AKT activation may be important in colitis development [28]. Studies have suggested that the tumorigenic function of CHI3L1 correlates with the aggressive behavior of tumor cells, which can be used as an independent molecular marker to predict poor prognosis in gastric cancer patients through activation of the AKT pathway [29,30]. CHI3L1 promotes AKT3 signal during intervertebral disc degeneration [31] and gastric cancer development [32]. In our in vitro study, consistent with the inhibitory effects of ebractenoid F on lung cancer cell growth, phosphorylated AKT expression was significantly decreased by ebractenoid F. Moreover, the combination of ebractenoid F and the AKT inhibitor further inhibited lung cancer cell growth and migration and induced apoptosis. Our previous study revealed higher levels of CHI3L1 expression and phosphorylated AKT in tissue samples in patients with lung cancer [21]. These results indicate that ebractenoid F inhibits lung cancer cell growth via the inhibition of AKT pathway-associated CHI3L1 signals. Based on these results, ebractenoid F may inhibit the growth of lung cancer cells through inhibition of the AKT pathway associated with CHI3L1 signaling. According to an in silico examination of the drug’s absorption, distribution, metabolism, and excretion, ebractenoid F is projected to have drug-like qualities; further, it showed low toxicity as per the predicted toxicology studies. Thus, ebractenoid F may serve as a promising anti-lung cancer agent.

## 4. Materials and Methods

### 4.1. Chemical Compounds

Ebractenoid F was isolated from the dried roots of Euphorbia fischeriana, as described previously [33]. The structure of ebractenoid F (Appendix A) was determined by the comparison of its spectroscopic data with those of published values [12]. 

### 4.2. Cell Culture and Transfection

A549 and H460 cells were obtained from the American Type Culture Collection (Manassas, VA, USA). The cells were cultured in RPMI 1640 media supplemented with 10% heat-inactivated fetal bovine serum (FBS), 100 g/mL penicillin, and 100 g/mL streptomycin. The cell cultures were maintained in a 37 °C incubator with a 5% CO_2_ humidified environment. According to the manufacturer’s instructions, the A549 and H460 cells were transiently transfected with pcDNA 3.1-CHI3L1 plasmid or CHI3L1 siRNA using Lipofectamine 3000 (for plasmid DNA) and RNAiMAX (for siRNA) reagent in Opti-MEM. We bought CHI3L1 siRNA from OriGene (Rockville, MA, USA).

### 4.3. Cell Viability Assay

The A549 and H460 lung cancer cells were plated in 96-well plates and then treated with ebractenoid F at 0, 17, 35, and 70 μM for 24 h. According to the manufacturer’s instructions, the MTT [3-(4,5-dimethylthiazol-2-yl)-2,5-diphenyltetrazolium bromide] assay (Sigma Aldrich, St. Louis, MO, USA) was used to determine the cells’ viability after treatment. Briefly, MTT (5 mg/mL) was added, and the plates were incubated at 37 °C for 4 h before 150 μL of dimethyl sulfoxide (DMSO) was added to each well. Finally, the absorbance of each well was read at a wavelength of 570 nm using a microplate reader. Experiments for each cell line were performed in triple experiments. The *IC*_50_ value was calculated using the MTT assay data.

### 4.4. Luciferase Activity Assay

CHI3L1 promoter dual-reporter (HPRM 12518-LvPG04) and negative control (NEG-LvPG04) lentiviral plasmid vectors were purchased from GeneCopoeia (Rockville, MD, USA). pcDNA3.1+6Myc-tagged CHI3L1 was cloned (Bionics). According to the manufacturer’s instructions, the A549 cells were plated in 12-well plates and transfected with a mixture of plasmid and Lipofectamine 3000 in Opti-MEM medium for 18 h to evaluate the transcriptional luciferase activity (Invitrogen, Carlsbad, CA, USA). After an appropriate time had passed since the transfection, the medium was harvested to measure the luciferase activity for the analysis of dual-luciferase activity. Luciferase activity was measured by using the Secrete-Pair™ Dual Luminescence Assay kit (GeneCopoeia). The data were read using a luminometer (WinGlow, Bad Wildbad, Germany).

### 4.5. Cell Migration Assay

For the wound-healing assay, the A549 and H460 cells were seeded in an SPLScar Block cell culture dish (#201935, SPL, Pocheon, Republic of Korea). This was composed of a 500 um thick wall to artificially generate a cell-free gap. The cells were incubated for 24 h after the culture medium was immediately replaced with RPMI containing ebractenoid F. Using an Olympus light microscope, migrated cell pictures were observed and analyzed using ImageJ software (NIH).

For the trans-well assay, the A549 and H460 cells were seeded on upper chamber inserts (8 μm pore trans-well; Corning Inc., New York, NY, USA). The ebractenoid F-treated lung cancer cells were plated at 2.0 × 10^4^ cells per well and incubated at 37 °C, 5% CO_2_, in a humidified incubator. The cells were then fixed with 4% formaldehyde for 5 min and permeated with 100% methanol for 15 min before being stained with 0.1% crystal violet for 20 min. In the upper chamber, non-migrated cells were removed with a cotton swab. Using an Olympus light microscope, migrated cell pictures were observed and analyzed using ImageJ software (NIH).

### 4.6. Cell Cycle Assay

To examine the cell cycle arrest of lung cancer cells, ebractenoid F-treated A549 and H460 cells were seeded on a 12-well plate, harvested by trypsin, pelleted, re-suspended in PBS, and fixed in 70% ethanol. At least 1–2 h before the flow cytometric analysis, the cells were washed in PBS and stained with 500 μL of PI/RNase (Propidium Iodide) solution. The flow cytometric analysis was performed with the flow cytometry system (FACS Calibur-S System; BD Biosciences, San Jose, CA, USA).

### 4.7. Evaluation of Apoptotic Cell Death

According to the manufacturer’s instructions, the DeadEnd^TM^ Fluorometric TUNEL System (Promega, Madison, Wisconsin, WI, USA) was used for the TUNEL assay to detect apoptotic cells. Lung cancer cells (2 × 10^4^ cells/well) were cultured on 8-chamber slides after being were treated with ebractenoid F. The cells were washed with PBS and fixed by 4% paraformaldehyde in PBS for 20 min at room temperature. They were then permeabilized by 0.1% Triton X-100 in PBS for 5 min at room temperature. The slides were incubated with mounting media for fluorescence-containing DAPI (Vector Laboratories, Inc., Burlingame, CA, USA) for 15 min, at room temperature and in the dark. After this, the cells were visualized using the ZEISS Axio Observer fluorescence microscope system (Carl Zeiss, Oberkochen, Germany). Digital images were analyzed using ZEN 2.1 software (Carl Zeiss).

### 4.8. Docking Experiment

The SwissDock (http://www.swissdock.ch, 4 June 2021) web service based on EADock DSS was used to predict the interaction of CHI3L1 with ebractenoid F. For molecular docking studies, the crystal structure of the CHI3L1 available at the PDB (Protein Data Bank) database was retrieved and prepared. The PDB file format of the CHI3L1 structure (PDB ID 1NWR) was downloaded from the PDB database (https://www.rcsb.org/, 4 June 2021). Before the docking study, the water molecules were deleted, and polar hydrogens were added to the structures. The Discovery studio (ver. 2021, DASSAULT, FRANCE) was used to prepare the removal of water molecules and addition of hydrogen molecules, and for building the structure of ebractenoid F. The detailed experimental method was carried out according to the following references [34]

### 4.9. Pull-Down Assay

Ebractenoid F was conjugated with cyanogen bromide-activated Sepharose 4B and epoxy-activated Sepharose 6B (Sigma-Aldrich, Burlington, Massachusetts, MA, USA). In brief, coupling buffer (0.1 M of NaHCO_3_ and 0.5 M of NaCl, pH 10) was used to dissolve 1 mg of ebractenoid F. The CNBr-activated Sepharose 4B and epoxy-activated Sepharose 6B were swelled and washed in 1 mM of HCl through a sintered glass filter, then washed with the coupling buffer. Beads were incubated in the coupling buffer containing ebractenoid F overnight at 4 °C. Three cycles of alternating pH wash buffers (buffer 1, 0.1 M of acetate and 0.5 M of NaCl, pH 4.0; buffer 2, 0.1 M of Tris-HCl and 0.5 M of NaCl, pH 8.0) were used to wash the ebractenoid F-conjugated Sepharose 4B and 6B. Ebractenoid F-conjugated beads were then equilibrated with a binding buffer (0.05 M of Tris-HCl and 0.15 M of NaCl, pH 7.5). The control unconjugated CNBr-activated Sepharose 4B and epoxy-activated Sepharose 6B beads were made as previously described. Ebractenoid F-conjugated Sepharose 4B and 6B were mixed with the lysate of myc-tagged human CHI3L1 transfected cells and incubated at 4 °C for 24 h. The beads were then washed with TBST three times. SDS loading buffer was used to elute the bound proteins. The proteins were then resolved by SDS-PAGE, followed by immunoblotting with antibodies against anti-myc (dilution 1:2000, #AE070, Abclonal, Wuhan, China).

### 4.10. Cycloheximide Chase Assay

The cycloheximide chase assay was performed as described previously [35] to assess CHI3L1 protein stability. The cells were pre-treated with ebractenoid F for 12 h and treated with cycloheximide (1 μg/mL) (Sigma-Aldrich, Burlington, Massachusetts, MA, USA) for the indicated time point, which was analyzed by Western blotting. The band intensities of CHI3L1 were calculated by ImageJ software (National Institutes of Health, Bethesda, MA, USA), and the values were normalized against β-actin for each sample.

### 4.11. Western Blotting

Western blotting was performed as described previously [21]. Lung cancer cells were harvested with lysis buffer (50 mM of Tris-HCl (pH7.6), 0.1% Triton X-100, 0.25 mM of NaCl, and 2 mM of EDTA with protease inhibitor and protease inhibitor) and lysed by 30 min of incubation on ice. The cell lysate was centrifuged at 12,000× *g* for 30 min at 4 °C. In total, 10 μg of proteins was subjected to SDS-PAGE for separation and then transferred to PVDF membrane. The membranes were blocked with 5% BSA in PBS and incubated with specific primary antibodies overnight. Following HRP-conjugated secondary antibodies incubation, the desired proteins were detected using ECL substrate (#WBKLS0500, Millipore, Billerica, MA) and visualized by using a FUSION Solo S chemiluminescence detection system (Vilber Lourmat, Collégien, France).

### 4.12. Statistical Analysis

The statistical analysis was performed as described previously [21]. Statistical analyses were carried out using the GraphPad Prism 5 software. All error bars reported are the standard deviation (SD) unless otherwise indicated. Pairwise comparisons were performed using a Student’s *t*-test. Multiple comparisons were performed using a one-way analysis of variance, followed by Tukey’s tests. Differences between groups were considered significant at *p*-values of < 0.05.

## Figures and Tables

**Figure 1 molecules-28-00329-f001:**
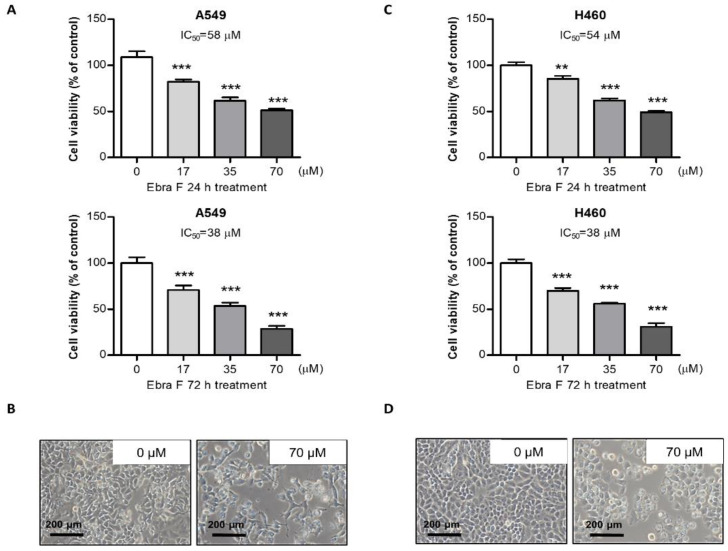
Effect of ebractenoid F on lung cancer cell growth. The morphology of lung cancer cells treated with ebractenoid F. Morphological changes of A549 and H460 lung cancer cells were observed under a phase-contrast microscope (**A**,**C**). Concentration-dependent effect of ebractenoid F on A549 and H460 cells observed using the cell viability assay (MTT assay) after 24 h and 72 h (**B**,**D**). The data were expressed as the mean ± S.D. of three experiments. **, *p* < 0.01; ***, *p* < 0.001; n.s. (not statistically significant) indicates statistically significant differences from the control group.

**Figure 2 molecules-28-00329-f002:**
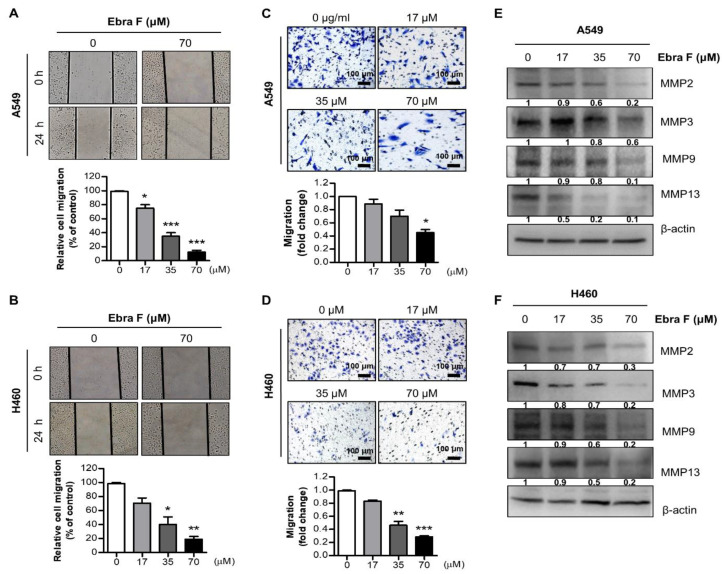
Effect of ebractenoid F on lung cancer cell migration. A wound-healing assay was performed in lung cancer cells after treatment with ebractenoid F for 24 h (**A**,**B**). A549 and H460 cells were seeded onto 6-well plates, and the confluent cell layer was scratched with 100 µL tips. Then, the cells were treated with different concentrations of ebractenoid F. The wound closure was monitored under microscope after 24 h and photographs were captured at 0 h and 24 h after wound generation. Then, trans-well assays were performed using A549 and H460 cells (**C**,**D**). Cells were plated into the gelatin-coated upper chamber of a 24-well format trans-well plate and treated with different concentrations of ebractenoid F. After 24 h of incubation, the migrated cells were stained and photographed under a microscope. The figures representatively show only the results of treatment with ebractenoid F concentrations of 0 and 70 µM. The data were expressed as the mean ± S.D. of three experiments. *, *p* ≤ 0.05; **, *p* < 0.01; ***, *p* < 0.001; n.s. (not statistically significant) indicates statistically significant differences from the control group. The expression levels of migration-associated proteins were determined by Western blot analysis using antibodies against MMP2, MMP3, MMP9, MMP13, and β-actin (internal control). Each band is representative of three experiments (**E**,**F**).

**Figure 3 molecules-28-00329-f003:**
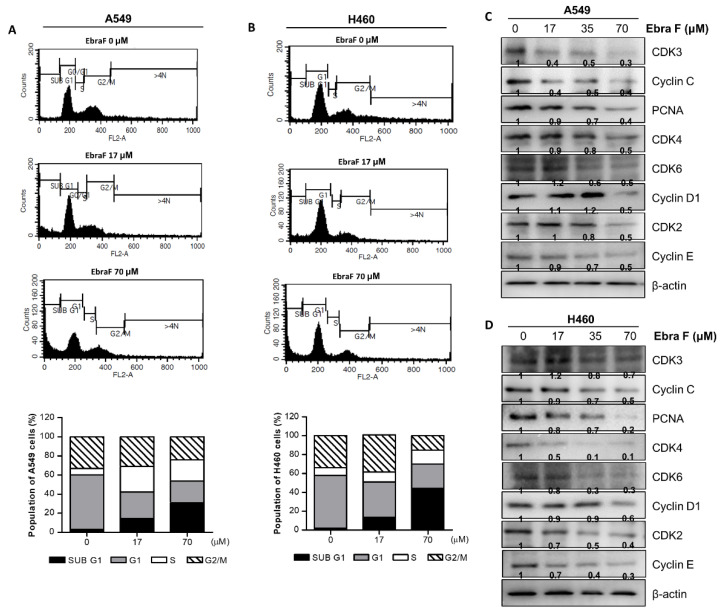
Effect of ebractenoid F on the cell cycle in lung cancer. Flow cytometric analysis of ebractenoid F-treated A549 and H460 cells (**A**,**B**). Ebractenoid F induced cell cycle arrest at the sub-G1 phase in lung cancer cells. Flow cytometry was used to determine the distribution of ebractenoid F-treated lung cancer cells in different phases of the cell cycle. Statistical analysis of the cell cycle distribution of lung cancer cells after 24 h of treatment with different concentrations of ebractenoid F. The expression levels of cell cycle-related proteins were determined by Western blot analysis using antibodies against CDK3, Cyclin C, PCNA, CDK4, Cyclin D1, CDK2, Cyclin E, and β-actin (internal control) (**C**,**D**). All data are expressed as the mean ±SD of three independent experiments.

**Figure 4 molecules-28-00329-f004:**
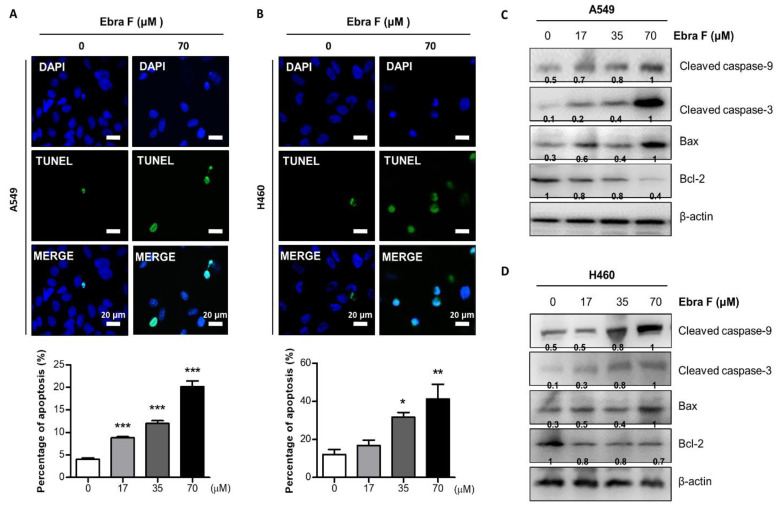
Effect of ebractenoid F on lung cancer cell death. The lung cancer cells were treated with ebractenoid F for 24 h, and then labeled with DAPI and TUNEL stains. Fluorescent microscopy images of TUNEL assay. DAPI staining (blue) was used to identify cell nucleus; TUNLE-positive cells are in green. The apoptotic index was determined as the DAPI-stained TUNEL-positive cell number/total DAPI stained cell number (magnification, 100×) (**A**,**B**). The values represent the means ± S.D. of three experiments. *, *p* ≤ 0.05; **, *p* < 0.01; ***, *p* < 0.001; n.s. (not statistically significant) indicates statistically significant differences from the control cells. The expression levels of proteins associated with the intrinsic apoptosis pathway were determined by Western blot analysis using antibodies against cleaved caspase-3, -9, Bax, Bcl-2, and β-actin (internal control). Each band is representative of three experiments (**C**,**D**).

**Figure 5 molecules-28-00329-f005:**
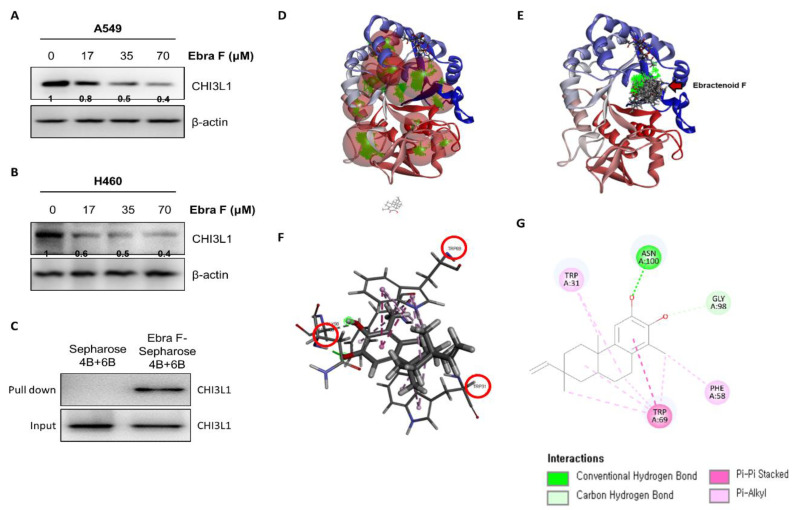
Interaction and molecular binding between ebractenoid F and CHI3L1. Differences in the expression levels of CHI3L1 in lung cancer cells after treatment with different concentrations of ebractenoid F (**A**,**B**). Pull-down assay showing the interaction between ebractenoid F and CHI3L1 (**C**). Ebractenoid F was conjugated with epoxy-activated Sepharose 4B and 6B. The part marked with red spheres: 1 to 12 cavities of CHI3L1 are represented (**D**). Docking model of ebractenoid F with CHI3L1. Ebractenoid F could be docked in these 12 cavities with the highest binding potential, and docking was possible with 14 different forms (**E**). Receptor–ligand interaction: 3D model (**F**). The binding energy delta G of the most energetically stable model among the 14 docking forms of ebractenoid F was calculated to be −7.66 kcal/mol. Receptor–ligand interaction: 2D model (**G**). The amino acids that play an important role in the binding of this docking are tryptophan31, tryptophan69, phenylalanine58, asparagine100, and glycine98.

**Figure 6 molecules-28-00329-f006:**
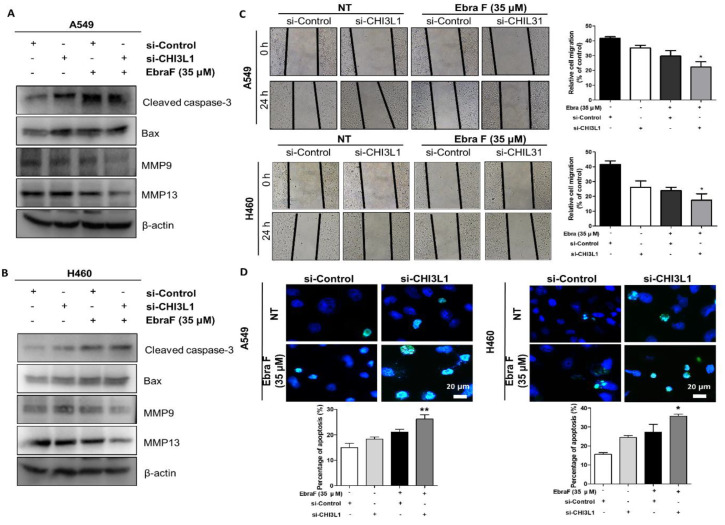
Combination effects of ebractenoid F and CHI3L1 siRNA on cell growth, migration, and apoptosis, as well as related protein expression. The lung cancer cells were transfected with CHI3L1 siRNA for 24 h and then treated with ebractenoid F for 24 h. The expression of migration and apoptosis-related proteins were detected by Western blot analysis in A549 and H460 cells (**A**,**B**). Wound healing assay was performed in the same way as previously mentioned (**C**). TUNEL assay was performed in the same way as previously mentioned (**D**). The values represent the means ± S.D. of three experiments. *, *p* ≤ 0.05; **, *p* < 0.01; n.s. (not statistically significant) indicates statistically significant differences from the control cells.

**Figure 7 molecules-28-00329-f007:**
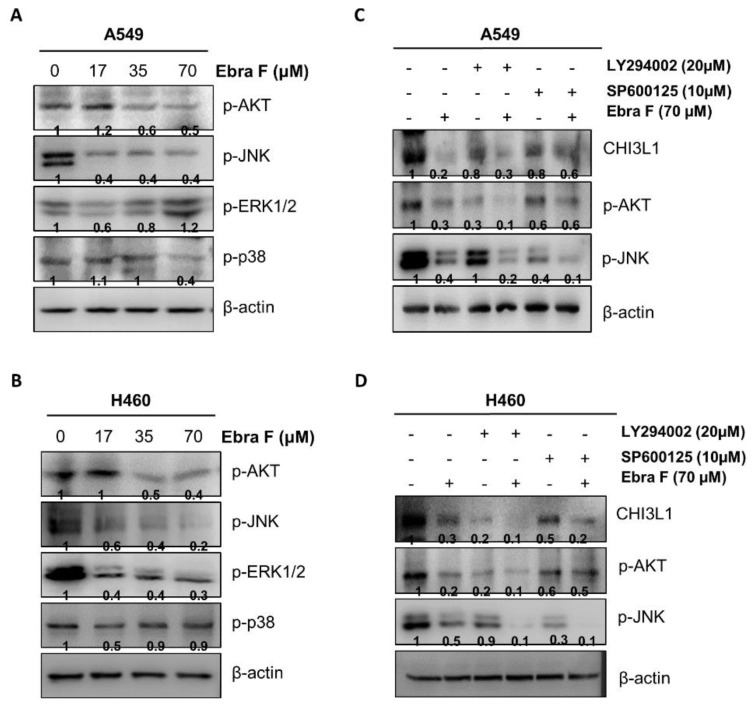
Combination effect of ebractenoid F with AKT inhibitor LY294002 or JNK inhibitor SP600125 on the expression of cell growth signal-associated protein. The position of CHI3L1 was important as an AKT activation site. After treating lung cancer cells with different concentration of ebractenoid F, the levels of cell growth signaling-associated proteins were determined by Western blot analysis using antibodies against p-AKT, p-JNK, p-ERK, p-p38, and β-actin (internal control) (**A**,**B**). A549 and H460 cells were pretreated with AKT inhibitor LY294002 (20 µM) and JNK inhibitor SP600125 (10 µM) for 30 min, the media were removed, and the cells were treated with ebractenoid F for 24 h (**C**,**D**). The expressions of CHI3L1, p-AKT, p-JNK, and β-actin (internal control) were detected by Western blotting using specific antibodies against each of these proteins.

**Figure 8 molecules-28-00329-f008:**
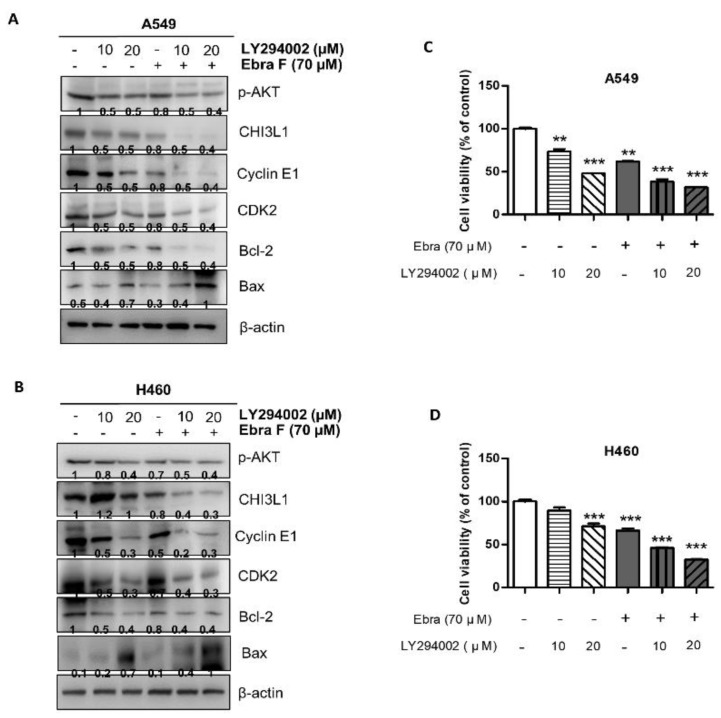
Combination treatment with ebractenoid F and AKT inhibitor LY294002 on cell growth, apoptosis, as well as expression of cell cycle and death signal protein. After the lung cancer cells were pretreated with AKT inhibitor LY294002, the expression levels of cell cycle (Cyclin E and CDK2) and apoptosis-related proteins (Bcl-2 and Bax) were detected by Western blot analysis (**A**,**B**). β-actin protein was used as an internal control. Each band is representative of three experiments. The concentration-dependent effects of ebractenoid F and LY294002 on A549 and H460 cells after 24 h of treatment were assessed via the cell viability assay (MTT assay) (**C**,**D**). The data were expressed as the mean ± S.D. of three experiments. **, *p* < 0.01; ***, *p* < 0.001; n.s. (not statistically significant) indicates statistically significant differences from the control group. Treatment of lung cancer cells with the combination of ebractenoid F and LY294002 for 24 h was first performed by their labeling with the DAPI and TUNEL stains. Fluorescent microscopy images of TUNEL assay. DAPI staining (blue) was used to identify cell nuclei; TUNLE-positive cells are in green. The apoptotic index was determined as the DAPI-stained TUNEL-positive cell number/total DAPI stained cell number (magnification, 100×) (**E**,**F**). The values represent the means ± S.D. of three experiments. **, *p* < 0.01; ***, *p* < 0.001; n.s. (not statistically significant) indicates statistically significant differences from the control cells.

**Table 1 molecules-28-00329-t001:** ADME/toxicity and druggability of Ebractenoid F.

Characteristic	Activity Value
LogP	5.82/Very lipophilic
M.W	286.40/Good
H-bond donors	2/Good
H-bond acceptors	2/Good
Number of rotatable bonds	1/Good
Number of rings	3/Good
Lipinski	Moderate (1)
Lead-like	Moderate (1)
Solubility	6.27/Highly insoluble
P-gp substrates	Non-substrate undefined
CYP1A2 Inhibitor	0.44/Undefined
CYP2C9 Inhibitor	0.48/Undefined
CYP2C19 Inhibitor	0.49/Undefined
CYP2D6 Inhibitor	0.44/Undefined
CYP3A4 Inhibitor	0.47/Undefined
Ames	Negative, Class 5
hERG	0.03/Non-Inhibitor
Caco-2	Highly permeability
PPB	94.72/Extensive bound
CNS	Penetrant
HIA	100/Highly absorbed
Metabolic stability	Undefined

## Data Availability

Not applicable.

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
