# Peer review of "A Natural CHI3L1—Targeting Compound, Ebractenoid F, Inhibits Lung Cancer Cell Growth and Migration and Induces Apoptosis by Blocking CHI3L1/AKT Signals"

_molecules, 2022, doi:10.3390/molecules28010329_

Round 1

Reviewer 1 Report

The organization and analysis of the reported references in this manuscript were of high quality. I think it can be published in this journal after minor revision.

1) To be complete, the authors should enrich the related illustration about the article on ebractenoid F in “Introduction”.

2) More aggressive and well-designed combination drug therapies, which exhibit a better additive or synergistic effects against lung cancer via a multi-pronged attack on different targets, are a promising strategy. The authors use the term “combination treatment” when describing ebractenoid F. A brief mention of the role “combination treatment” playing in the treatment of lung cancer is needed. (using the relevant references, such as DOI: 10.1038/nrclinonc.2014.225; DOI: 10.3390/biomedicines9060689; DOI: 10.1016/j.jconrel.2017.05.023 and DOI: 10.1039/d2nj03231g).

3) The sentence starting on Page 11, line 297-299, “Lung tumor growth and metastasis were also effectively controlled through STAT6-dependent M2 polarization inhibition by anti-CHI3L1 antibodies (Yu et al., 2021).” should be vetted;

On Page 11, line 323-334, “Several natural compounds inhibited lung cancer cell growth through the inhibition of the AKT pathway…..Moreover, erianthridin has shown to suppresses non- small-cell lung cancer cell metastasis via the inhibition of AKT signaling pathway [32].” It is not appropriate to the contents in the conclusion section.

Reviewer 2 Report

The manuscript entitled "A natural CHI3L1-targeting compound ebractenoid F inhibits lung cancer cell growth and migration and induce apoptosis by blocking of CHI3L1/AKT signals" contains scientifically important information, but needs a few corrections as listed below:

1. Title: use the term "induces"

2. Figure 1B and 1D lack scale and magnification

3. Line 93: insert “with” in-between “treated various”

4. Figures 2A and 2B lack clarity

5. Line 175: “We then examed” spelling mistake with ”examined

6. Line 315 – 346: Check the font size

Reviewer 3 Report

In this manuscript, Hong et al investigated the anti-cancer effect of a natural compound, ebractenoid F. Using two lung cancer-derived cell lines, the authors demonstrated that ebractenoid F could inhibit cell growth and migration, arrest cell cycle, and increase apoptosis. Since ebractenoid F was selected based on its inhibitory effect on CHI3L1 expression, the authors based their mechanistic study on the interaction between ebractenoid F and CHI3L1. They found that ebractenoid F can directly bind to CHI3L1 protein. They also found that ebractenoid F can suppress AKT signaling pathway.

Both the phenotypic studies (growth, migration, cell cycle, apoptosis), and the mechanistic studies (CHI3L1 expression, CHI3L1 binding, connection to AKT signaling), were well performed. However, those two lines of study are not correctly integrated. According to the data presented, the effect of ebractenoid F on cell growth, migration, cell cycle, apoptosis, CHI3L1 expression and AKT signaling can all be individually confirmed. But there is no evidence showing that the effect of ebractenoid F on cellular phenotypes is mediated by its effect on CHI3L1 and AKT signaling.

Some specific comments:

1.    All the data on the combination effects, such as those presented in Figure 6, 7, 8, does not provide any evidence on the underlying mechanism as interpreted by the authors. For example, in Figure 6, the fact that “The combined treatment of CHI3L1 siRNA and ebractenoid F more significantly reduced the expression of migration proteins compared to the treatment of ebractenoid F or CHI3L1 siRNA alone”, can neither confirm nor reject the hypothesis that the effect of ebractenoid F on cell migration is mediated by CHI3L1. Moreover, the fact that adding CHI3L1 siRNA can further inhibit cell migration, as shown in the manuscript, strongly suggests that CHI3L1 is not the only mediator of ebractenoid F’s effect on cell migration. To gain mechanistic insight, I suggest the authors perform rescue assay, instead of combination assay, to pinpoint the regulatory cascade.

2.    The images for wound healing assays presented in Figure 2A, 2B and 6C are not very convincing. The proportion of migrated cells does not seem to differ much based on the current image. Please consider replacing with more representative images. Also, for Figure 2A and 2B, why not show images from 20 ug/ml treatment?

3.    Section 2.5 has the wrong title, this section has nothing to do with CHI3L1, it is all about apoptosis.

4.  Section 2.3 and 2.6 have the same title.   
